# LLM PRUNING AND DISTILLATION IN PRACTICE

## ABSTRACT

Structured pruning with knowledge distillation is a potent combination for obtaining small language models (SLMs) with significantly fewer training tokens and compute resources compared to training from scratch. In this work, we investigate how this strategy can be effectively applied in instances where access to the the original pretraining dataset is restricted. We introduce a new *teacher correction* phase before distillation which lets the teacher model adjust to our specific data distribution using a lightweight fine-tuning phase. We apply this strategy to compress the Mistral NeMo 12B and Llama 3.1 8B models to 8B and 4B parameters, respectively, using pruning and distillation. We explore two distinct pruning strategies: (1) depth pruning and (2) joint hidden/attention/MLP (width) pruning, and evaluate the results on common benchmarks from the LM Evaluation Harness. The models are then aligned with NeMo Aligner and further tested for instruction following, role-play, math, coding and function calling capabilities. This approach produces the state-of-the-art Mistral-NeMo-Compressed-8B (MN-COMPRESSED-8B for brevity) model from Mistral NeMo 12B, and a compelling 4B model from Llama 3.1 8B.

## 1 INTRODUCTION

LLM providers often train an entire family of models from scratch, each with a different size (number of parameters, e.g. Llama 3.1 with 8B, 70B, and 405B parameters (Dubey & et al, 2024)); this is done to aid users targeting different deployment scales, sizes and compute budgets. However, training multiple billion-plus parameter models from scratch is extremely time-, data- and resource-intensive. Recent work has demonstrated the effectiveness of combining weight pruning with knowledge distillation to significantly reduce the cost of training LLM model families Muralidharan et al. (2024). Here, only the biggest model in the family is trained from scratch; other models are obtained by successively pruning the bigger model(s) and then performing knowledge distillation Hinton et al. (2015) to recover the accuracy of

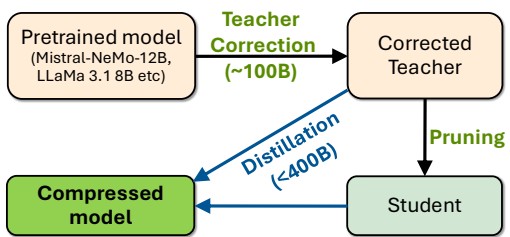

Figure 1: High-level overview of our proposed pruning and distillation approach. The total number of tokens used for each step is indicated in parentheses.

pruned models. While highly effective, this line of work assumes access to the original pretraining dataset for the distillation phase. With a growing number of frontier LLMs (including open ones) being trained on private, proprietary datasets Dubey & et al (2024); Team et al. (2024), this assumption often fails to hold.

In this work, we adapt the original Minitron compression recipe (Muralidharan et al., 2024) along two directions: (1) we introduce a new *teacher correction* phase for adapting the teacher (unpruned) model to our own data distribution, thus removing any need to access the original pretraining dataset, and (2) we introduce a new and more effective downstream task-based saliency criteria for depth pruning. We successfully apply

| Benchmarks(shots) | Gemma2 2B* | Minitron 4B | Llama-3.1-Compressed 4B-Depth | 4B-Width | Gemma 7B | Mistral 7B | Llama 3.1 8B | MN-Compressed 8B | Mistral NeMo 12B-Base | 12B-FT |
|---|---|---|---|---|---|---|---|---|---|---|
| Total Params | 2.6B | 4.2B | 4.5B | 4.5B | 8.5B | 7.3B | 8B | 8.4B | 12.2B | 12.2B |
| Non-Emb. Params | 2B | 2.6B | 3.7B | 3.7B | 7.7B | 7B | 7B | 7.3B | 10.9B | 10.9B |
| Training Tokens | 2T | **94B** | **94B** | **94B** | 6T | 8T | 15T | **380B** | - | +0.1T |
| Winogrande(5) | 70.9 | **74.0** | 72.1 | 73.5 | 78 | 78.5 | 77.3 | **80.4** | 82.2 | 82.7 |
| Arc_challenge(25) | 55.4 | 50.9 | 52.6 | **55.6** | 61 | 60.3 | 57.9 | **64.4** | 65.1 | 62.3 |
| MMLU(5) | 51.3 | 58.6 | 58.7 | **60.5** | 64 | 64.1 | 65.3 | **69.5** | 69.0 | 70.1 |
| Hellaswag(10) | 73.0 | 75.0 | 73.2 | **76.1** | 82 | **83.2** | 81.8 | 83.0 | 85.2 | 85.3 |
| GSM8k(5) | 23.9 | 24.1 | 16.8 | **41.2** | 50 | 37.0 | 48.6 | **58.5** | 56.4 | 55.7 |
| Truthfulqa(0) | - | **42.9** | 38.2 | **42.9** | 45 | 42.6 | 45.0 | **47.6** | 49.8 | 48.3 |
| XLSum en(20%) (3) | - | **29.5** | 27.2 | 28.7 | 17 | 4.8 | 30.0 | **32.0** | 33.4 | 31.9 |
| MBPP(0) | 29.0 | 28.2 | 30.7 | **32.4** | 39 | 38.8 | 42.3 | **43.8** | 42.6 | 47.9 |
| HumanEval(n=20)(0) | 20.1 | **23.3** | - | - | 32.0 | 28.7 | 24.8 | **36.2** | 23.8 | 23.8 |

Table 1: Accuracy numbers for our MN-COMPRESSED-8B and LLAMA 3.1-COMPRESSED-4B models. We compare our models to similarly-sized SoTA open models on a variety of common language modeling benchmarks. All evaluations are conducted by us, except entries marked with * (taken from corresponding papers).

| Benchmarks | Phi-2 2.7B | Gemma2 2B | Qwen2 1.5B | Minitron 4B | Llama-3.1-Compressed 4B-Depth | 4B-Width | LLama 3.1 8B | MN-Compressed 8B |
|---|---|---|---|---|---|---|---|---|
| MT-Bench (GPT4-Turbo) | 5.14 | **7.44** | 5.49 | 6.46 | 6.16 | 6.78 | 7.78 | **7.86** |
| MMLU (5) | 56.8 | 56.9 | 55.6 | 59.3 | 60.4 | **61.1** | 69.4* | **70.4** |
| GSM8K (0) | 19.9 | 52.2 | 27.2 | 65.1 | 72.5 | **75.2** | 83.8 | **87.1** |
| GPQA (0) | 28.8 | 25.9 | 28.1 | 29.5 | 23.2 | **30.1** | 30.4* | **31.5** |
| HumanEval (0) | **47.6*** | 45.1 | 47.0* | 39.6 | 33.5 | 36.2 | **72.6** | 71.3 |
| MBPP (0) | 55.0* | 50.4 | 51.9* | 54.2 | 56.9 | **57.4** | **72.8*** | 72.5 |
| IFEval | 44.0 | 64.5 | 39.8 | 75.3 | 71.0 | **76.6** | 80.4* | **84.4** |
| BFCLv2 (Live) | 38.7 | 40.2 | 39.9 | 53.1 | 56.3 | **59.6** | 44.3 | **67.6** |

Table 2: Accuracy numbers for instruction tuned models on a variety of benchmarks. All evaluations are conducted by us, except entries marked with * (taken from corresponding papers). Best of each section in **bold**. For IFEval, we report the average of prompt and instruction across loose and strict evaluations. For BFCLv2, we report live accuracy only.

our updated compression strategy to two state-of-the-art models: Llama 3.1 8B Dubey & et al (2024) and Mistral NeMo 12B team (2024), compressing them down to 4B and 8B parameters, respectively. For Llama 3.1 8B, we produce two distinct compressed models: (1) LLAMA 3.1-COMPRESSED-4B-Width (pruning only the width axes), and (2) LLAMA 3.1-COMPRESSED-4B-Depth (pruning depth only). Figure 1 provides a high-level overview of our approach.

Tables 1 and 2 provide a summary of our results: our compression strategy yields a state-of-the-art 8B model (MN-COMPRESSED-8B) which outperforms all similarly-sized models across the board on common language modeling benchmarks. Our LLAMA 3.1-COMPRESSED-4B models (both depth and width-pruned variants) also exhibit strong accuracy compared to the teacher Llama 3.1 8B model and the previous-generation Minitron-4B model Muralidharan et al. (2024); among the two variants, the width-pruned variant achieves better overall accuracy than the depth-pruned one. In terms of runtime inference performance measured using TensorRT-LLM, the LLAMA 3.1-COMPRESSED-4B models provide an average speedup of 2.7× and 1.8× for the depth and width pruned variants, respectively, compared to the original Llama 3.1 8B model.

This paper makes the following key contributions:

1. Introduces a new step before pruning and distillation named teacher correction which helps the teacher model adapt to a user's own data distribution.

2. Presents a new and improved depth pruning saliency metric based on downstream task accuracy.

3. Successfully applies the new pruning recipe to the Llama 3.1 8B and Mistral NeMo 12B models to produce three state-of-the-art compressed models; the new recipe continues to enjoy the significant cost and training token reductions demonstrated in earlier pruning+distillation work Muralidharan et al. (2024).

## 2 METHODOLOGY

A high-level overview of our approach is illustrated in Figure 1. Here, the teacher model undergoes a lightweight adjustment phase on the target dataset to be used for distillation - we refer to this step as *teacher correction*. Next, pruning is applied to compress the model, following which distillation is used to recover model accuracy.

### 2.1 TEACHER CORRECTION

Distillation is an effective technique to condense knowledge from a more accurate teacher model to improve a less accurate student model Hinton et al. (2015) Muralidharan et al. (2024). Typically, knowledge is distilled using the same dataset the teacher model was trained on. In cases where access to the original training data is restricted, we notice from our experiments that the teacher model provides sub-optimal guidance if a different dataset is used to distill the knowledge. We hypothesize this is due to the change in distribution of sub-word tokens across the original dataset the teacher model was trained on vs. the dataset being distilled on. To this end, we propose a novel teacher correction phase (illustrated in Figure 2), where we perform a lightweight (~100B tokens) fine-tuning of the teacher model to adapt to the new distillation dataset. We demonstrate in Section 5 (Figure 3 in particular) that this procedure significantly improves the guidance resulting in a more accurate student model. We also explore correcting the teacher in parallel to distillation, and demonstrate that this performs on par with using guidance from a fully corrected teacher.

### 2.2 PRUNING

Weight pruning is a powerful and well-known technique for reducing model size. In this paper, we focus on structured pruning, where blocks (or channels) of nonzero elements are removed at once from model weights; examples of structured pruning techniques include neuron, attention head, convolutional filter, and depth pruning Xia et al. (2023); Ashkboos et al. (2023); Men et al. (2024); Kim et al. (2024). In this paper, we follow the pruning recipe outlined in Minitron Muralidharan et al. (2024): we start the pruning process by first computing the importance of each layer, neuron, head, and embedding dimension. We then sort these importance scores to compute a corresponding importance ranking.

**Importance Estimation** We use a purely activation-based importance estimation strategy that simultaneously computes sensitivity information for all the axes we consider (depth, neuron, head, and embedding channel) using a small calibration dataset and only forward propagation passes. We consider depth pruning as a special case and do not combine it with compressing other dimensions. We compute the importance of each head, neuron and embedding channel by examining the activations produced by the multi-head attention (MHA), multi-layer perceptron (MLP) and LayerNorm layers, respectively. We use a small calibration dataset (1024 samples) for this purpose.

**Layer Importance** For depth pruning, we consider two distinct metrics for evaluating layer importance: (1) LM validation loss/PPL, and (2) accuracy on the downstream task. We do not consider the Block Importance (BI) metric Men et al. (2024) as it was recently shown to under-perform the validation loss/PPL metric Muralidharan et al. (2024). For ranking, we simply remove a single or a block of contiguous layers and compute its effect on each metric; this serves as the "importance" or sensitivity of the layer/layer block. Based on our empirical analysis (see Section 4; specifically, Figures 7 and 8), we use the Winogrande metric (Sak-

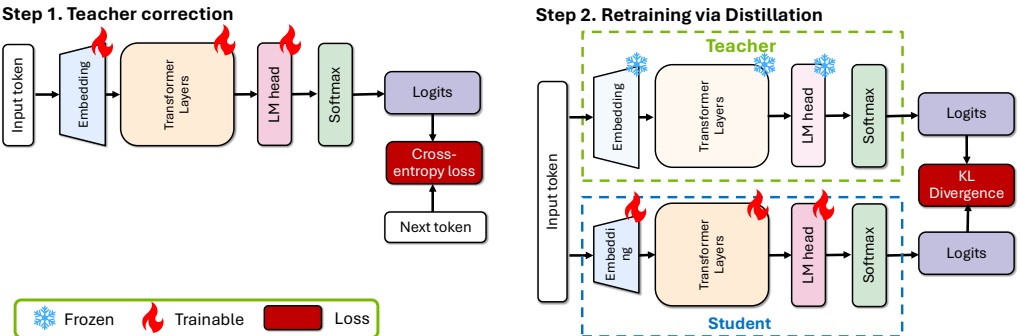

Figure 2: Overview of distillation: if/when the original training data is unavailable, a lightweight fine-tuning of the original model on the distillation dataset is recommended, to be used as a teacher. Distillation is then performed by minimizing KL divergence on the logits of the teacher and the pruned student model.

aguchi et al., 2021) to prune sets of contiguous layers. This pruning strategy evolved from two important observations: (1) LM validation loss/PPL-based layer importance fails to produce the most accurate pruned model(s) on downstream tasks, and (2) dropping contiguous layers is better than individual, as also observed in Gromov et al. (2024).

**Model Trimming** Following Muralidharan et al. (2024), for a given architecture configuration, we first rank the elements of each axis according to the computed importance and perform trimming of the corresponding weight matrices directly. For neuron and head pruning, we trim MLP and MHA layer weights, respectively. In the case of embedding channels, we trim the embedding dimension of the weight matrices in MLP, MHA, and LayerNorm layers. The original approach (Muralidharan et al. (2024)) uses Neural Architecture Search (NAS) to find the best architecture; in this work, we skip this step and instead utilize the network architecture-related learnings from the original paper.

### 2.3 RETRAINING WITH DISTILLATION

We use the term retraining to refer to the accuracy recovery process post pruning. In this work, we explore two retraining strategies: (1) conventional training, leveraging ground truth labels, and (2) knowledge distillation using supervision from the unpruned model (teacher). Knowledge Distillation (KD) Hinton et al. (2015) involves transfer of knowledge from a larger or more complex model called the teacher to a smaller/simpler model called the student. The knowledge transfer is achieved by having the student model mimic the output and/or the intermediate states of the teacher model. In our case, the uncompressed and pruned models correspond to the teacher and student, respectively. Following the best practices outlined in the Minitron work Muralidharan et al. (2024), we use forward KL Divergence loss Kullback & Leibler (1951) on the teacher and student logits only. This is illustrated in Figure 2.

## 3 TRAINING DETAILS

### 3.1 PRE-TRAINING

Llama 3.1 8B (Dubey & et al, 2024) and Mistral NeMo 12B (team, 2024) are pretrained on different proprietary datasets, which we do not have access to. According to the Llama 3.1 tech report Dubey & et al

(2024), the 8B model is pretrained on 15T tokens. We start with the corresponding Base models that are openly available on Hugging Face.

**Dataset** We use a proprietary dataset consisting of high-quality pretraining data (which, to our knowledge, does not overlap with the ones used to train Llama 3.1 and Mistral NeMo) for all our pruning and distillation experiments.

## 3.2 TEACHER CORRECTION

Using the original Mistral NeMo 12B or Llama 3.1 8B models directly as a teacher performs sub-optimally on our dataset. To counter this, we apply teacher correction, as described in Section 2, to both models with $\sim 100B$ tokens. Since the goal is to adapt the teacher model to the distillation dataset, we use 120 steps of warm-up and low learning rates: one-fifth the peak learning rate, identical batch size, minimum learning rate and decay schedule the original model was trained on. We notice that the correction process has a minor effect on the teacher model's accuracy on downstream tasks, with some tasks improving and some degrading as shown in Table 1. We hypothesize this to be an artifact of the dataset used for fine-tuning. Optimizing this process further by using fewer than ~100B tokens, lighter fine-tuning such as LoRA Hu et al. (2021) or tuning layer normalization Ba et al. (2016) parameters alone would be an interesting topic for future work.

## 3.3 PRUNING

Our pruning recipe is based on the best practices outlined in the Minitron paper Muralidharan et al. (2024) and is described in Section 2. Specifically, for width pruning, we (1) use `l2-norm` and `mean` as the aggregation functions across the batch and sequence dimensions, respectively, and (2) perform single-shot pruning, avoiding iterative approaches. For depth pruning, as described in Section 2, we follow the observations from Gromov et al. Gromov et al. (2024) and drop a continuous subgroup of layers that results in the least accuracy drop on Winogrande Sakaguchi et al. (2021). In this work, we skip the lightweight neural architecture search (NAS) phase, and go with a manual architecture configuration for both LLAMA 3.1-COMPRESSED-4B and MN-COMPRESSED-8B. The architectures we come up with are inspired by the Minitron-4B and Minitron-8B models Muralidharan et al. (2024), and are detailed in Table 3.

## 3.4 DISTILLATION

As described in Section 2, we opt for logit-only distillation, minimizing the forward KL Divergence Kullback & Leibler (1951) loss across the teacher and student probabilities, and ignore the LM cross-entropy loss altogether. Here, the un-pruned and pruned models correspond to the teacher and student, respectively. We

| | LLaMa-3.1-Compressed-4B | | MN-Compressed |
| --- | --- | --- | --- |
| | Width | Depth | 8B |
| Total params | 4.5B | 4.5B | 8.4B |
| Non-Emb params | 3.7B | 3.5B | 7.3B |
| Hidden size | 3072 | 4096 | 4096 |
| Vocabulary | 128256 | 128256 | 131072 |
| MLP hidden dim | 9216 | 14336 | 11520 |
| Depth | 32 | 16 | 40 |
| Attention groups | 8 | 8 | 8 |
| Query heads | 32 | 32 | 32 |
| Head dimension | 128 | 128 | 128 |

Table 3: Architecture details of our compressed models.

| | Llama-3.1-Compressed-4B | MN-Compressed 8B |
| --- | --- | --- |
| Peak learning rate | 1e-4 | 1e-4 |
| Min learning rate | 1e-5 | 4.5e-7 |
| Warm-up steps | 40 steps | 60 steps |
| LR decay schedule | Cosine | Cosine |
| Global batch size | 1152 | 768 |
| Context length | 8192 | 8192 |
| Total tokens | 94B | 380B |

Table 4: Hyperparameters used during distillation-based retraining.

use the hyperparameters listed in Table 4 during distillation. We use 32 NVIDIA DGX H100 nodes for our training jobs.

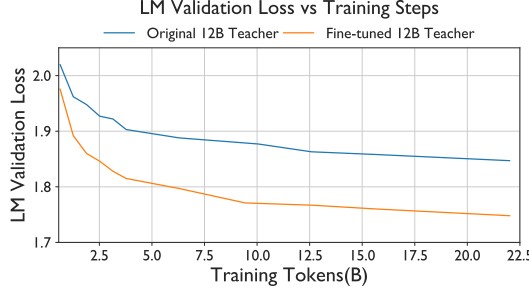 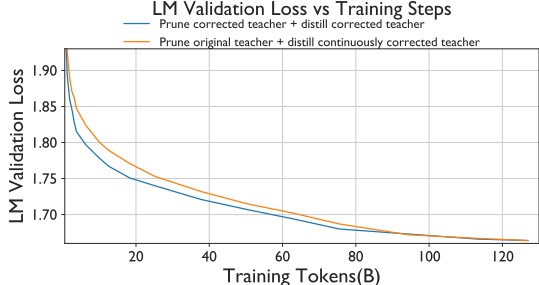

Figure 3: Training convergence plot for the MN-COMPRESSED-8B student model. We compare supervision from the original teacher and the corrected teacher.

Figure 4: Training convergence plot for the MN-COMPRESSED-8B student model. We compare (1) pruning and distilling the corrected teacher with (2) pruning the original (uncorrected) teacher and distilling from a continuously corrected teacher. We notice that teacher correction can be performed in parallel with distillation.

### 3.5 INSTRUCTION TUNING

To evaluate the instruction-following capabilities of our distilled models, we perform alignment using NeMo-Aligner Shen et al. (2024). We follow the same recipe for all our models by first applying math and code supervised fine-tuning (SFT) followed by instruction SFT and then two rounds of Reward-aware Preference Optimization (RPO) Nvidia et al. (2024).

## 4 ANALYSIS

We perform a series of ablation studies to better understand the effects of distillation, teacher correction, and our new depth-pruning saliency metric. We report our findings in this section.

**Teacher Correction** We first compare the effects of teacher correction on the MN-COMPRESSED-8B model in Figure 3; here, we notice the clear benefits of performing teacher correction w.r.t. distilling directly from an uncorrected teacher. Next, we compare two approaches for teacher correction: (1) pruning and distilling the corrected teacher, and (2) pruning the original (uncorrected) teacher and distilling from a continuously corrected teacher. The results in Figure 4 suggest that teacher correction can be performed in parallel with distillation to recover accuracy of the pruned student model.

**Pruning and Distillation** Figure 5 demonstrates the orthogonal benefits of pruning and distillation over random initialization and conventional fine-tuning, respectively. We compare (1) random weight initialization and distillation, (2) random pruning and distillation, where weights are pruned randomly ignoring the importance scores, (3) our proposed pruning with typical cross entropy based LM loss training and (4) our proposed pruning with distillation-based retraining. We notice that pruning results in a significantly better starting point compared to random initialization, and distillation-based training outperforms conventional training methods. Overall, our approach requires significantly fewer training tokens (up to $40\times$; 380B instead of 15T tokens) to produce the state-of-the-art MN-COMPRESSED-8B model.

**Width vs. Depth Pruning**   Figure 6 shows the training curve of LLAMA 3.1-COMPRESSED-4B pruned for width vs. depth. We notice that width pruning results in a lower initial loss and consistently outperforms the depth-pruned model, despite both variants having the same number of parameters.

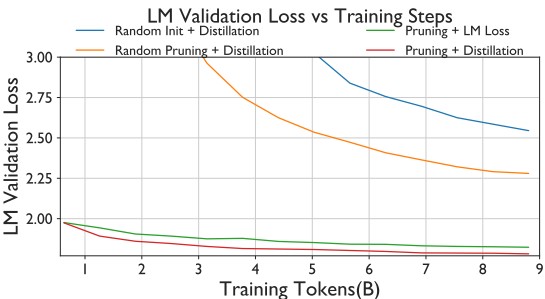 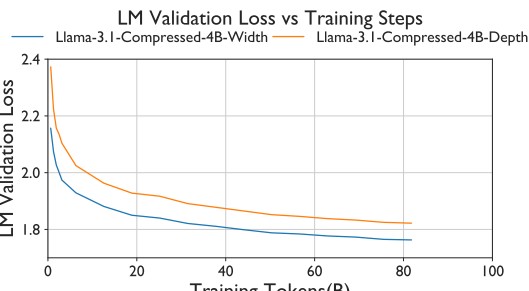

Figure 5: Training convergence plot for the MN-COMPRESSED-8B model. We compare (a) random initialization with distillation, (b) randomly pruned weights with distillation, (c) pruning with standard LM loss, and (d) our pipeline with pruning and distillation. This plot shows the benefits of pruning and distillation over random initialization and conventional finetuning, respectively.

Figure 6: Convergence plots for the width-pruned and depth-pruned versions of Llama 3.1 8B to 4B compressed models. Width pruning consistently outperforms depth pruning for a given parameter budget.

**Depth Pruning Metrics**   By examining how LM validation loss increases as contiguous blocks of layers are removed (Figure 7), we observe that the layers at the beginning and end are the most important. The figure indicates that removing non-contiguous layers can result in even better LM validation loss (the dashed line). However, we notice this observation does not necessarily hold when evaluating downstream task performance: specifically, Figure 8 shows that dropping 16 layers selected based on per-layer importance (Men et al. (2024); Siddiqui et al. (2024)) yields a random Winogrande accuracy of 0.5, while removing layers 16 to 31 continuously (Gromov et al. (2024)) results in an accuracy of 0.595. The gap holds during distillation-based retraining and we opt for the latter approach in this work.

## 5 EVALUATION

**Benchmarks**   following Touvron et al. (2023), we evaluate our compressed base and aligned models on a series of downstream tasks, namely MMLU Hendrycks et al. (2021), HumanEval Chen et al. (2021b) for Python code generation, MBPP Austin et al. (2021) and GSM8K Cobbe et al. (2021). We also evaluate the base models on several question-answering datasets for common-sense reasoning: Arc-C Clark et al. (2018), HellaSwag Zellers et al. (2019), TruthfulQA Lin et al. (2022), WinoGrande Sakaguchi et al. (2021), and XL-Sum English Hasan et al. (2021) for summarization. The instruction tuned models are further evaluated for question-answering, function calling, instruction following and multiturn conversations on GPQA Rein et al. (2023), BFCL Yan et al. (2024), IFEval Zhou et al. (2023) and MT-Bench (GPT4-Turbo) Wang et al. (2024), respectively. Note that this MT-Bench is a corrected version of the original MT-Bench Zheng et al. (2023).

For base models, accuracy is reported with the following evaluations settings: 5-shot on MMLU, 5-shot on Winogrande, 25-shot on ARC-Challenge, 10-shot on HellaSwag, 0-shot on 20% of XL-Sum and average pass@1 scores for HumanEval and MBPP. For pass@1 scores we use a temperature of 0.2 and nucleus

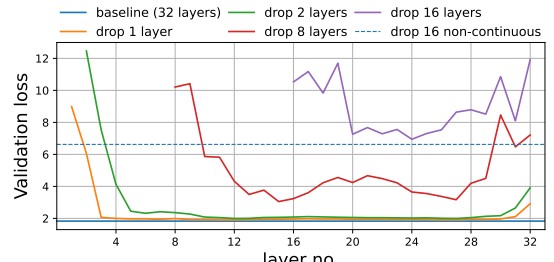

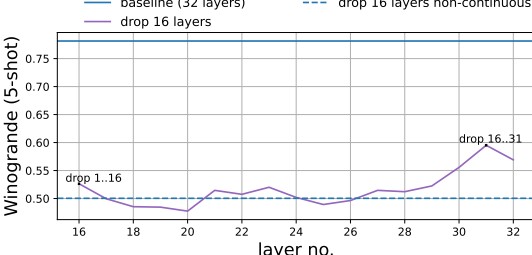

Figure 7: LM loss value on validation set after removing 1, 2, 8 or 16 contiguous layers from Llama 3.1 8B. The purple line at layer no. 16 indicates the LM loss if we dropped the first 16 layers. Layer no. 17 indicates the LM loss if we leave the first layer intact and drop layers 2 to 17. The dashed line corresponds to LM loss value when removing 16 non-contiguous layers least increasing the loss.

Figure 8: Accuracy on the Winogrande task when removing 16 contiguous layers from Llama 3.1 8B. Layer no. 17 indicates the accuracy if we leave the first layer intact and drop layers 2 to 17. The dashed line corresponds to the accuracy when removing 16 non-contiguous layers that increasing the loss by the least amount.

sampling Holtzman et al. (2019) with top-p = 0.95. For aligned models we use 0 shot and greedy sampling if applicable.

## 5.1 BASE MODELS

Base model evaluation results are shown in Table 1. Compared to similarly-sized models, MN-COMPRESSED-8B demonstrates superior accuracy across the board, outperforming the recent Llama 3.1 8B model using $40\times$ fewer training tokens (380B vs. 15T). Similarly, the LLAMA 3.1-COMPRESSED-4B models perform favorably compared to the teacher Llama 3.1 8B model using $150\times$ fewer training tokens (94B vs. 15T); our pruned Llama models also outperform the Minitron 4B model Muralidharan et al. (2024). We note from Table 1 that the width-pruned variant outperforms the depth-pruned one. These results clearly demonstrate the advantages of our methodology: state-of-the-art accuracy coupled with an order of magnitude improvement in training efficiency.

## 5.2 INSTRUCT MODELS

The accuracy of the instruction-tuned model variants are shown in Table 2. Our aligned models outperform similarly sized variants on most evaluated benchmarks with the exception of HumanEval Chen et al. (2021a) and MBPP Austin et al. (2021). Additionally, LLAMA 3.1-COMPRESSED-4B lags behind Gemma2 on MT-Bench Zheng et al. (2023). Nevertheless, our aligned models are consistently better on MMLU Hendrycks et al. (2021), GSM8K Cobbe et al. (2021), GPQA Rein et al. (2023), IFEval Zhou et al. (2023) and BF-CLv2 Yan et al. (2024). This demonstrates the strong capabilities of our model.

## 5.3 RUNTIME PERFORMANCE ANALYSIS

To evaluate runtime performance, we optimize the Llama 3.1 8B and LLAMA 3.1-COMPRESSED-4B variants with NVIDIA TensorRT-LLM, an open-source toolkit for optimized LLM inference.

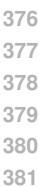

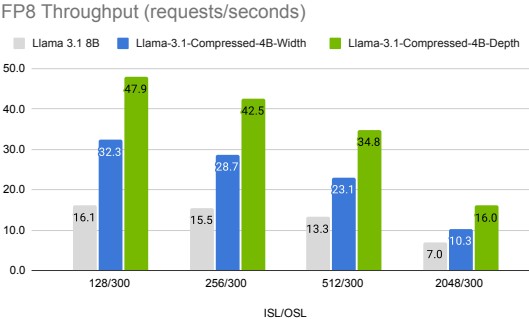

Figure 9: TensorRT-LLM FP8 throughput comparison for the LLAMA 3.1-COMPRESSED-4B models with the Llama 3.1 8B model w.r.t. increasing input and output sequence lengths.

Figure 9 shows the throughput in requests per second for the various models in FP8 precision obtained on a single H100 80 GB GPU. Different use cases are represented by increasing input sequence length/output sequence length (ISL/OSL) combinations, at a batch size of 32 and 64 for the 8B-12B models and the 4B models respectively. The smaller memory footprint of the 4B model allows for larger batches. We notice that LLAMA 3.1-COMPRESSED-4B (Depth) is fastest, achieving an average throughput improvement of $2.7\times$ over Llama 3.1 8B; the width-pruned variant achieves an average throughput improvement of $1.8\times$ over Llama 3.1 8B. Compared to BF16, we notice that FP8 delivers a performance boost of $1.4\times$.

## 6 INSIGHTS

In this section, we summarize some interesting and surprising observations based on our evaluation.

**General**

1. Teacher correction is crucial for distillation to work optimally on a new, unseen dataset. Fine-tuning the teacher with the dataset used for distillation in this manner yields over a 6% reduction in LM validation loss. Teacher correction doesn't affect the optimality of pruning and can even be performed in parallel with distillation.

2. In line with the Minitron paper's observations, we require a order of magnitude fewer tokens (380B vs 15T) to achieve state-of-the-art accuracy post pruning with distillation.

3. For width pruning, we achieve stronger accuracy by retaining attention heads and pruning the other dimensions (MLP intermediate dimension, embedding channels).

**Mistral NeMo 12B to MN-COMPRESSED-8B**

1. Our compressed model outperforms the teacher on two benchmarks, GSM8k and HumanEval after pruning and distillation: GSM8k increases from 55.7% to 58.5% and HumanEval increases from 23.8% to 36.2%. This improvement is likely influenced by the dataset. However, retraining is performed using the distillation loss alone.

**Llama 3.1 8B to LLAMA 3.1-COMPRESSED-4B**

1. Width pruning delivers better accuracy with MMLU at 60.5%, while depth pruning yields 58.7%, for Llama 3.1 compression.

2. Reasoning ability for base variants appears to be impacted significantly for the depth pruned version, with GSM8K accuracy at 16.8% compared to 41.24% for the width pruned version. However, the gap reduces with instruct tuning.

3. Depth pruning boosts throughput, achieving $2.7\times$ speedup over Llama-3.1 8B, while width pruning provides $1.7\times$ speedup.

4. For depth pruning, we observe that dropping contiguous layers from the model is more effective than using non-contiguous, importance-based pruning.

## 7 RELATED WORK

Structured pruning is a well-studied area, with a recent crop of papers specifically focusing on LLM compression. We can broadly classify these works into ones that target depth (layers) (Men et al., 2024; Yang et al., 2024; Kim et al., 2024) and ones that reduce width (hidden dimension, attention heads, MLP intermediate size, etc.) (Xia et al., 2023; Dery et al., 2024; Ashkboos et al., 2023; Ma et al., 2023); a small subset targets both axes Muralidharan et al. (2024); Xia et al. (2023). Among recent papers, we choose to adopt and extend the Minitron work Muralidharan et al. (2024) for several key reasons: first, to the best of our knowledge, it provides the first systematic pruning recipe that targets both width and depth axes using a low-cost importance estimation criteria (based on forward-propagation passes only); many other approaches (eg: gradient-based ones) are significantly costlier in terms of training compute and thus less practical for LLMs. Secondly, it achieves state-of-the-art performance compared to other similar compression methods on modern LLMs.

Teacher correction appears to be a novel area of exploration. Recent work focuses on adapting the teacher to (1) address the capacity gap with respect to the student, where the teacher is fine-tuned based on knowledge distillation constraints Huang et al. (2022), and (2) address batch-norm statistics when using out-of-distribution data (different downstream tasks) for distillation with convolution based models on image tasks Szatkowski et al. (2023). To the best of our knowledge, ours is the first work specifically targeted at LLMs that adapts the teacher to provide optimal guidance on a dataset not identical to the original dataset the teacher model was initially trained on.

## 8 CONCLUSIONS

This paper has presented a novel strategy for applying pruning and distillation to models when access to the original pretraining dataset is restricted. Teacher correction, which performs lightweight finetuning of the teacher model on the target dataset significantly improves accuracy in this setting. This paper has also presented a novel saliency metric for layers that improves depth-pruning accuracy over existing approaches. Using this new pruning recipe, we produce a state-of-the-art 8B model (MN-COMPRESSED-8B) from Mistral NeMo 12B and a set of compelling 4B models (LLAMA 3.1-COMPRESSED-4B) from Llama 3.1 8B.

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
