# OpenReview forum: "LLM Pruning and Distillation in Practice"
_ICLR.cc/2025/Conference — Submitted to ICLR 2025_

### Official Review · Reviewer_r4i4 · 2024-10-24

**Soundness:** 2
**Presentation:** 1
**Contribution:** 2
**Rating:** 3
**Confidence:** 5

**Summary:**

The paper explores the practical application of structured pruning and knowledge distillation to compress large language models (LLMs) efficiently. The main goal is to create smaller models with fewer parameters while maintaining performance, even when the original pretraining dataset is unavailable. To achieve this, authors proposed

**Teacher Correction**: A novel phase where the teacher model is fine-tuned with a small amount of new data to adapt to the specific distribution of the target dataset. This improves the distillation process when the original training data is inaccessible.

**Pruning Strategies**: Authors explored two pruning strategies, including depth pruning and width pruning with their proposed importance measurement for each component.

**Strengths:**

1. The authors provide their base model on Hugging Face, and based on this, the results seem promising.

2. Teacher correction can improve the effectiveness of distillation when the original data is unavailable, as demonstrated with Llama 3.1 8B and Mistral NeMo 12B.

3. The authors propose a new method for measuring importance in pruning, but there seems to be a lack of comparison with other methods for evaluating LLM importance.

**Weaknesses:**

1. This paper does not follow the ICLR 2025 style guidelines, which could be a serious reason for desk rejection.

2. The overall writing is somewhat unclear. I believe the writing could be improved, and some details in the paper also need enhancement, including the proper use of \citep and \citet, as well as capitalization at the beginning of sentences.

3. The reasoning behind the improvement from teacher correction is not very clear. Even though it is a crucial component of the work, it seems to only apply to a single case. I recommend that the authors provide more analysis or experiments demonstrating the effectiveness of teacher correction across different models or datasets. This would help establish whether the improvement is generalizable or limited to a specific case.

4. Overall, the main components of the paper—distillation and pruning—appear to overlap significantly with previous works, particularly Sheared LLaMA, which first proposed efficient training using distillation and pruning. However, the paper does not cite this work.

**Questions:**

1. There are several methods for measuring the importance of layers, including SLEB, Shortened LLaMA, and ShortGPT. For a stronger publication, I suggest that the authors include a comparative analysis section that directly compares their proposed importance measurement method with these existing methods on a common set of metrics or benchmarks.

2. Is there any empirical validation for the authors' argument: "We hypothesize this is due to the change in the distribution of sub-word tokens across the original dataset the teacher model was trained on versus the dataset being distilled on"? I recommend that the authors provide quantitative evidence of the token distribution differences between datasets and demonstrate how these differences correlate with the effectiveness of teacher correction.

---

> ### Author Response · Authors · 2024-11-22
> **Author Rebuttal**
>
> We would like to thank the reviewer for their encouraging comments and insightful feedback. Please find our responses below:
>
> > **W1: ICLR 2025 style guidelines**
>
> We apologize for the confusion - we have followed the ICLR 2025 style guidelines to the best of our knowledge. We will re-verify the paper contents to ensure it meets all guidelines.
>
> > **W2: I believe the writing could be improved, and some details in the paper also need enhancement, including ...**
>
> Thank you for pointing this out. We will incorporate the suggested changes in the final version.
>
> > **W3: I recommend that the authors provide more analysis or experiments demonstrating the effectiveness of teacher correction across different models or datasets.**
>
> This is a relevant question. We validate the effectiveness of teacher correction by evaluating it on two distinct model families (Llama 3.1 and Mistral-NeMo, trained on different pretraining datasets). Figure 3 in the paper shows the effects of applying teacher correction on Mistral-NeMo 12B. In our internal experiments, we noticed a similar trend for the Llama 3.1 family as well.
>
> > **W4: Comparison to ShearedLLaMa.**
>
> Thank you for pointing this out. To help with this comparison, we have created the following table to highlight the differences between our approach and Sheared LLaMa. We will add the appropriate citations to ShearedLLaMa in the final version.
>
> |Criteria|Sheared LLaMa|Our Approach|Summary|
> |:---|:---|:---|:---|
> |Pruning importance estimation | Learnable mask on  embedding, MLP and attention heads. Mask learning uses 0.4B tokens and is 5x slower than standard LM training (see ShearedLLaMa paper). | Forward pass on a tiny number of samples (1024 in paper) to compute importance for embedding, MLP and attention heads at once. This adds negligible overhead. | An expensive gradient-based importance estimation strategy (Sheared-Llama) is not required to achieve strong accuracy for pruned models. A simpler, forward pass only approach (as in this work and Minitron) works well.|
> | Pruning dimensions | Depth, embedding, MLP, attention | Depth, embedding, MLP, attention | Both approaches support multiple width and depth axes.|
> |Retraining | Uses conventional finetuning | Uses knowledge distillation | Both papers find that retraining is required to recover accuracy loss on benchmarks. | We showcase the superiority of knowledge distillation over conventional training and recommend the former as the retraining approach.|
> |Multiple compressed models|Requires repeating the 5x slower mask learning process N times for producing N compressed models.|Each of the N models must also be finetuned.|Single importance estimation pass (negligible overhead) is sufficient for all N compressed models. Each of the N models must be distilled.|Our approach is significantly less costly when multiple compression targets are specified.|
>
> > **Q1: Comparisons to SLEB, Shortened LLaMA, and ShortGPT**
>
> The table below compares our approach to SLEB, Shortened LLaMa, SliceGPT and ShortGPT:
>
> |Model|HellaSwag|Winogrande|MMLU|Arc-c|
> |:---|---:|---:|---:|---:|
> |Shortened-Llama (4.5B)|59.4|59.3|NA|34.5|
> |ShortGPT (4.9B)|53.02|NA|43.96|NA|
> |SLEB (5.6B)|62.47|58.96|NA|33.02|
> |SliceGPT (4.9B)|50.27|NA|28.92|NA|
> |**Our depth pruned model (4.5B)**|**73.2**|**72.1**|**58.7**|**52.6**|
>
> > **Q2: I recommend that the authors provide quantitative evidence of the token distribution differences between datasets and demonstrate how these differences correlate with the effectiveness of teacher correction.**
>
> This is a great suggestion. Unfortunately, we have no access to the original datasets (Llama or Mistral-NeMo) to perform this comparative study across dataset distributions.

---

> > ### Comment · Reviewer_r4i4 · 2024-12-02
> >
> > Thanks for the authors' response. However, it appears that the manuscript has not been updated to address even basic style guidelines, and there are still areas in the writing that could be improved. Based on this, I believe the paper may not yet be ready for acceptance at the conference. As my concerns have not been fully addressed, I will maintain my score.

---

### Official Review · Reviewer_7ENF · 2024-11-03

**Soundness:** 3
**Presentation:** 3
**Contribution:** 2
**Rating:** 6
**Confidence:** 3

**Summary:**

This paper addresses an increasingly important challenge in the field of large language models (LLMs): how to effectively compress models when access to the original training data is restricted. The authors present a practical approach combining structured pruning and knowledge distillation, introducing several notable innovations in the process. The key contributions include a novel "teacher correction" phase for adapting the teacher model to target dataset distribution, an improved depth pruning saliency metric based on downstream task performance, and empirical validation through compression of Mistral NeMo 12B to 8B and Llama 3.1 8B to 4B parameters.

**Strengths:**

- The paper introduces teacher correction as an innovative solution for compressing models without original training data access - a common industry challenge with proprietary models. The approach is elegantly simple yet effective, requiring only ~100B tokens for adaptation and integrating smoothly into existing distillation pipelines. This practical focus makes the method immediately valuable for real-world applications.
- The results show impressive efficiency gains with the MN-COMPRESSED-8B achieving SOTA performance using 40× fewer training tokens, and LLAMA 3.1-COMPRESSED-4B showing competitive results with 150× fewer tokens. The method delivers significant speedups (2.7× for depth pruning, 1.8× for width pruning) while maintaining strong performance across diverse tasks including reasoning, coding, and instruction following.

**Weaknesses:**

- The performance degradation observed in the compressed models is substantial, especially considering the modest compression ratios achieved. When compressing Llama 3.1 8B to 4B parameters (only a 2x reduction), the MMLU performance drops notably from 65.3% to 60.5% with width pruning, or even lower to 58.7% with depth pruning. This performance drop is particularly concerning when viewed against recent developments in the field - for instance, MobileLLM-350M demonstrates that it's possible to achieve comparable performance to LLaMA-v2 7B in specific tasks with a model that's 20 times smaller. The fact that this paper shows significant performance degradation with just 2x compression, while requiring additional fine-tuning, makes the proposed approach less compelling for practical applications.

- The evaluation of the method's effectiveness is hampered by insufficient baseline comparisons. The authors should have compared their approach against a broader spectrum of existing compression techniques, including various pruning approaches, quantization methods.

- The paper's experimental scope is also notably limited. While the authors validate their approach on two recent and popular models (Mistral NeMo 12B and Llama 3.1 8B), they don't explore more aggressive compression scenarios such as creating sub-1B parameter models, which would be particularly valuable for resource-constrained deployments.

[MobileLLM] MobileLLM: Optimizing Sub-billion Parameter Language Models for On-Device Use Cases

**Questions:**

- Why Winogrande was chosen over other potential downstream tasks?
- How sensitive is the method to the choice of downstream task for depth pruning?

---

> ### Author Response · Authors · 2024-11-22
> **Author Rebuttal**
>
> We would like to thank the reviewer for their encouraging comments and insightful feedback. Please find our responses below:
>
> > **W1: Comparison to MobileLLM-350M**
>
> We thank the reviewer for pointing out the drop in accuracy. Though small models like MobileLLM-350M may shine on particular tasks, they still typically underperform when evaluated across multiple downstream tasks. In particular, MobileLLM-350M beats LLaMA-v2 7B on API function calling (a specific downstream task) but not other downstream tasks. On this particular task, we observe a similar trend where our Llama3.1-compressed-4B outperforms the original 8B model (~2x larger model, BFCL v2 with 59.6% vs 44.3%). From Tables 1 and 2, we observe that despite an aggressive 50% compression ratio, our compressed models retain competitive scores across the full suite of downstream tasks w.r.t models trained from scratch with an order of magnitude more training tokens.
>
> > **W2: The authors should have compared their approach against a broader spectrum of existing compression techniques, including various pruning approaches, quantization methods.**
>
> Table 4 in the Minitron work [1] compares Minitron models to previous state-of-the-art compression techniques. Our compressed 8B and 4B models outperform all these baselines (including Minitron) as shown in Tables 1 and 2.  We will add a detailed comparison to related work, including LLMPruner, SliceGPT, ShortGPT, etc. in the final version.
>
> > **W3: Sub-1B parameter models**
>
> This is a great suggestion. We plan to apply our strategy to further compress existing models into the sub-1B parameter range in the near future.
>
> > **Q1: Why Winogrande was chosen over other potential downstream tasks?**
>
> Winogrande is a good representative of commonsense reasoning tasks, especially for smaller models. We draw this conclusion based on our internal evaluations that indicate Winogrande doesn’t saturate at these model sizes and continues to scale. It also has the advantage that it can be computed quickly.
>
> > **Q2: How sensitive is the method to the choice of downstream task for depth pruning?**
>
> We observed that different types of tasks utilize different heads of the model, though the general pattern is similar. Also we noticed that generation tasks like GSM8K are more sensitive to depth pruning than common knowledge tasks. The selection of candidates for further distillation may depend on the downstream task; we leave this topic for further research that targets domain-dependent pruning approaches.
>
> **References**
> 1. Compact Language Models via Pruning and Knowledge Distillation, arXiv:2407.14679v2, 2024

---

### Official Review · Reviewer_2Myx · 2024-11-04

**Soundness:** 3
**Presentation:** 2
**Contribution:** 2
**Rating:** 6
**Confidence:** 4

**Summary:**

This paper studies the combination of structural pruning and knowledge distillation to obtain compressed and performant language models with higher throughput. The paper proposes a "teacher correction" to allow for lightweight finetuning of the teacher model (a llama-3.1-8B or mistral nemo 12B). The step is proposed with the goal of adapting the distribution of the teacher to the domain of the finetuning data before the distillation step to a smaller model. Furthermore, the paper uses different saliency metrics to compute importance of depth and intermediate mlp/attention/width dimension and sort the teacher in order of importance, before applying pruning. In addition, the paper also explores different saliency metrics for improved depth pruning. The pruned models are evaluated on a wide variety of commonsense, math and instruction tuning tasks.

**Strengths:**

Experiments
- The paper evaluates the proposed teacher correction scheme on different models (llama-3.1-8B and mistral nemo 12B) for a more thorough study of effectiveness of distillation (pre-corrected or continuously corrected)
- Detailed evaluation of gains from different components random student initialisation, importance sorting and knowledge distillation
- Exhaustive evaluation on different tasks

Writing
- Paper is written in an easy to understand manner in most parts

Originality and significance
- While most parts of the paper are similar to/derive from observations in [1], the empirical investigation into precise gains by each of the components is interesting and significant.

[1] Muralidharan, S., Sreenivas, S.T., Joshi, R., Chochowski, M., Patwary, M., Shoeybi, M., Catanzaro, B., Kautz, J. and Molchanov, P., 2024. Compact language models via pruning and knowledge distillation. arXiv preprint arXiv:2407.14679.

**Weaknesses:**

Originality and Significance
- A lot of this work is similar to  [1], including importance sorting, architecture selection procedure, distillation loss
- Domain adaptation of a teacher is also studied in [2] for BERT like models and I don't find the proposed teacher correction to be significantly different from the idea of adapting a teacher to a specific domain on interest
- The idea of dropping last contiguous layers (except the last layer) have been studied in [3] and the scheme chosen here is similar to the scheme chosen in the paper. Is my understanding correct and could the authors present the key differences/observations here?

Experiments
- Check questions section, I think the paper would benefit a lot from adding comparisons to similar small sized models, trained from scratch.

Clarity
- While the paper is written in an easy to understand manner in most parts, I am lacking a cohesive story of the overall approach of the paper. Could the authors provide an algorithm describing the final/best performing choices for pruning+distillation?
- Could you elaborate on the following points which are ambiguous/unclear : 167-168, what are the architecture related learnings, how was the student designed?  Line 133-134, which calibration set is used, does the domain of the calibration set matter? "continuously corrected teacher", is not defined properly in the paper, is the teacher periodically updated? line 312-313, what is the general scheme used for pruning different teachers, which contiguous layer indices are to be dropped?

Scaling to larger models
- As per my understanding the paper studies only full fine-tuning (FFT) of the teacher model, for teacher correction. However, this does not scale to larger models (eg: llama-3.1-70B) due to memory/compute constraints and hence cannot exploit larger/better teacher models.
- While the authors do mention the possible use of Parameter-Efficient-Fine-Tuning (PEFT) schemes like LoRA[4] and Galore[5], I think given the focus of the paper on large language models, generalisability of the observations in the paper for PEFT schemes on even larger language models, should be studied in detail.

Code and Reproducibility
- The models are finetuned and distilled on a proprietary dataset, how do the observations translate to public datasets eg: openwebtext or finetuning datasets like alpaca, commonsense170k, math10k?
- The paper does not release the code and experimental pipeline for their work and I encourage the authors to do this to increase the impact and adoptability of their work.

[1] Muralidharan, S., Sreenivas, S.T., Joshi, R., Chochowski, M., Patwary, M., Shoeybi, M., Catanzaro, B., Kautz, J. and Molchanov, P., 2024. Compact language models via pruning and knowledge distillation. arXiv preprint arXiv:2407.14679.

[2] Yao, Y., Huang, S., Wang, W., Dong, L. and Wei, F., 2021. Adapt-and-Distill: Developing Small, Fast and Effective Pretrained Language Models for Domains. Findings of the Association for Computational Linguistics: ACL-IJCNLP 2021.

[3] Gromov, A., Tirumala, K., Shapourian, H., Glorioso, P. and Roberts, D.A., 2024. The unreasonable ineffectiveness of the deeper layers. arXiv preprint arXiv:2403.17887.

[4] Hu, E.J., Shen, Y., Wallis, P., Allen-Zhu, Z., Li, Y., Wang, S., Wang, L. and Chen, W., 2021. Lora: Low-rank adaptation of large language models. arXiv preprint arXiv:2106.09685.

[5] Zhao, J., Zhang, Z., Chen, B., Wang, Z., Anandkumar, A. and Tian, Y., 2024. Galore: Memory-efficient llm training by gradient low-rank projection. arXiv preprint arXiv:2403.03507.

**Questions:**

- Given the newly released llama-3.2-1b and llama-3.2-3b and ministral-3b models, could the authors compare against these model families?
- I currently lack a concrete answer to the question : Does a (student) model pruned and distilled from a larger model outperform a (student) model of similar size trained from scratch on a pretraining dataset? Could the authors elaborate on their observations here?
- Have the authors tried to replace full-fine-tuning with PEFT techniques like LoRA[1] and Galore[2]? If yes do the observations about distillation and importance sorting hold there?
- Could the authors provide on-device latency gains using the pruned models eg: on a A100/H100?

I am willing to raise my score if the authors adequately respond to my questions and the weaknesses pointed out above.

[1] Hu, E.J., Shen, Y., Wallis, P., Allen-Zhu, Z., Li, Y., Wang, S., Wang, L. and Chen, W., 2021. Lora: Low-rank adaptation of large language models. arXiv preprint arXiv:2106.09685.

[2] Zhao, J., Zhang, Z., Chen, B., Wang, Z., Anandkumar, A. and Tian, Y., 2024. Galore: Memory-efficient llm training by gradient low-rank projection. arXiv preprint arXiv:2403.03507.

---

> ### Author Response · Authors · 2024-11-22
> **Author Rebuttal Part 1**
>
> We would like to thank the reviewer for their encouraging comments and insightful feedback. Please find our responses below:
>
> > **W1: Domain adaptation vs. teacher correction**
>
> We agree with the reviewer that teacher correction is similar in spirit to domain adaptation. We wanted to highlight that in this case, finetuning on the distillation dataset is necessary even though we aren’t explicitly changing domains.
>
> > **W2: dropping last contiguous layers (except the last layer) [3] vs. our approach**
>
> The main difference in our work is that we use drop in Winogrande accuracy as the metric to choose which contiguous layers have to be dropped. This results in a more accurate pruned model compared to using LM loss as the metric (as in [3]). The Table below compares the two approaches.
>
> |Method|Training tokens|LM loss|
> |:----|:----|:----|
> |Gromov [3]|8.4B|2.092|
> |Ours|8.4B|2.062|
> | | | |
>
> > **W3: I think the paper would benefit a lot from adding comparisons to similar small sized models, trained from scratch.**
>
> Table 1 in the paper compares our models to various other similarly-sized community models trained from scratch (eg: Llama 3.1, Gemma/Gemma2, Mistral, etc.).
>
> > **W4: Could the authors provide an algorithm describing the final/best performing choices for pruning+distillation?**
>
> Our final approach for obtaining pruned+distilled models can be summarized as follows:
> * If no access to the original dataset, perform teacher correction using ~100B tokens from the distillation dataset.
> * Compute importance rankings for depth and width axes on corrected teacher. For width, use L2-norm and mean aggregation across the batch and sequence dimensions, respectively (as evaluated in [1]); for depth, find contiguous layers that least affect Winogrande accuracy.
> * Use the computed rankings to prune the corrected teacher along width and/or depth axes.
> * Distill knowledge from corrected teacher to pruned student model using logit only loss and forward KLDiv loss.
>
> > **W5: Could you elaborate on the following points which are ambiguous/unclear ...**
>
> **Lines 167-168**
> Based on the original Minitron paper, we focused on retaining network depth (except for Llama3.1-Depth) and attention heads. The remaining axes were pruned in proportion to the parameter budget with an emphasis on retaining hidden dimension size.
>
> **Lines 133-134**
> The 1024 calibration data samples are drawn randomly from the full distillation data corpus. We did not experiment with calibration data drawn from different domains, but we hypothesize that this will change the importance rankings proportional to how different the two data domains are.
>
> **Lines 312-313**
> Based on our observations, for depth based pruning, we prune a set of contiguous layers that results in the least drop in Winogrande accuracy.
>
> **Continuously corrected teacher**
> * In this setting, the (1) teacher correction and (2) distillation from the corrected teacher, are run in parallel.
> * The teacher model/checkpoint used in (2) is getting updated periodically through (1).
> * Continuous teacher correction is described in lines 268-273, with results presented in Figure 4.
>
> > **W6: Full fine-tuning vs. PEFT**
>
> Full fine-tuning uses the same resources as model pretraining and has been shown to scale to models well over the sizes we consider in this work. We agree that parameter-efficient fine-tuning would be a more resource-efficient way to perform teacher correction. We leave this exploration to future work. We also note that correction is only necessary when access to original training data is restricted.
>
> We agree that using PEFT for teacher correction (and potentially even distillation) would be an interesting avenue for future research.
>
> > **W7:  how do the observations translate to public datasets eg: openwebtext...**
>
> We have not evaluated our method on public datasets; however, we expect the general trends to remain the same. This would be another interesting avenue for future work.
>
> > **W8: Code release**
>
> We plan to release the code and experimental pipeline in the coming weeks, pending internal legal review. Model weights will also be released on HF with a permissive license. If required, we can also upload our checkpoints immediately to HF under an anonymous ID.
>
> **Responses to remaining questions provided in the follow-up comment.**

---

> > ### Author Response · Authors · 2024-11-22
> > **Author Rebuttal Part 2**
> >
> > > **Q1: Comparisons to Llama3.2 3B and Ministral 3B**
> >
> > Thank you for the suggestion. We have added a comparison to Llama-3.2 3B and Ministral 3B in the tables below. After submission, we were also able to compress the Mistral-NeMo-12B model to the 4B parameter range (MN-Compressed-4B). We include this model in our updated tables below. Our models remain competitive w.r.t. these newer models.
> >
> > **Base Models**
> >
> > ||Llama-3.1-Compressed 4B-Depth|Llama-3.1-Compressed 4B-Width|MN-Compressed 4B-Width|Llama 3.2 3B|Ministral 3B|
> > |:---|---:|---:|---:|---:|---:|
> > |Total Params|4.5B|4.5B|4.5B|3.2B|~3B|
> > |Training Tokens|94B|94B|94B|9T|NA|
> > |Winogrande(5)|72.1|73.5|**75.5**|59.6|72.7|
> > |Arc_challenge(25) opencompass|NA|**78.8**|NA|69.1|NA|
> > |MMLU(5)|58.7|60.5|**63.7**|58|60.9|
> >
> > ||MN-Compressed 8B|Ministral 8B|
> > |:---|---:|---:|
> > |Total Params|8.4B|8.02B|
> > | | | |
> > |Training Tokens|380B|NA|
> > | | | |
> > |Winogrande(5)|**80.4**|75.3|
> > |MMLU(5)|**69.5**|65|
> > |GSM8k(5)|58.5|**64.5**|
> > |HumanEval(n=20)(0)|**36.2**|34.8|
> >
> > **Aligned Models**
> >
> > |Benchmarks|Llama-3.1-Compressed 4B-Depth|Llama-3.1-Compressed 4B-Width|MN-Compressed 4B-Width|Llama 3.2 3B|Ministral 3B|
> > |:---|---:|---:|---:|---:|---:|
> > |MMLU(5)|61.2|59.9|**63.9**|63.4|NA|
> > |GSM8k(0)|71.1|79.8|**82.8**|77.7*|NA|
> > |GPQA (0)|32.6|30.4|29.7|**32.8**|NA|
> > |HumanEval (0)|42.7|47.0|60.4|NA|**77.4**|
> > |MBPP (0)|60.3|65.1|**71.2**|NA|67.7|
> > |IFEval|66.8|79.5|**80.1**|77.4|NA|
> > | | | | |* 8-shot CoT| |
> >
> > |Benchmarks|MN-Compressed 8B|Ministral 8B|
> > |:---|---:|---:|
> > |HumanEval (0)|71.3|**76.8**|
> > |MBPP (0)|**72.5**|70|
> >
> > > **Q2: Does a (student) model pruned and distilled from a larger model outperform a (student) model of similar size trained from scratch on a pretraining dataset?**
> >
> > Yes, as shown in Table 1 of the paper, our compressed models outperform similarly-sized models trained from scratch. For example, the MN-Compressed-8B model distilled on <400B tokens outperforms llama 3.1 8B trained on 15T tokens on all tasks.
> >
> > > **Q3: PEFT techniques**
> >
> > Please refer to our answer on PEFT above. For importance sorting, we only perform forward-propagation passes.
> >
> > > **Q4: On-device latency gains**
> >
> > Figure 9 in the paper provides the throughput improvements obtained by compressing the Llama-3.1 8B model (w.r.t. varying input and output sequence lengths).

---

> > > ### Comment · Reviewer_2Myx · 2024-11-25
> > >
> > > I thank the authors for addressing my questions and for the additional experiments. Though, I still think that the main ideas proposed in the paper eg: domain adaptation, importance sorting have been seen in previous literature, I do think the paper does a thorough job in investigating teacher correction. I am increasing my score to 6.

---

### Meta-Review · Area_Chair_NXiM · 2024-12-23

**Metareview:**

This paper tackles an increasingly relevant challenge in the field of large language models (LLMs): compressing models effectively when access to the original training data is restricted. The key contributions include a novel "teacher correction" phase to adapt the teacher model to the target dataset distribution, an enhanced depth pruning saliency metric based on downstream task performance, and empirical validation demonstrated through compressing Mistral NeMo 12B to 8B and Llama 3.1 8B to 4B parameters.

The strengths of the paper lie in its focus on an important research question and its demonstrated efficiency improvements. However, the work exhibits significant weaknesses, including limited technical novelty and non-compliance with the ICLR style guidelines.

Given these limitations, I recommend rejecting this submission.

**Additional Comments On Reviewer Discussion:**

During the discussion period, Reviewers ZMyx and r4i4 actively engaged with the authors.

After carefully reviewing the concerns raised by the reviewers, I found that the authors failed to adequately address several critical issues.

A central concern raised by all reviewers is the lack of technical novelty in the proposed method. Specifically, reviewers, including ZMyx, unanimously agreed that the method is incremental to prior works such as Minitron and Sheared LLaMA, though the empirical evaluations provided are valuable. Additionally, the concept of "teacher correction" bears a strong resemblance to domain adaptation techniques, yet no comparisons to related works or baselines were presented. Most importantly, as highlighted by Reviewer r4i4, the submission does not fully adhere to the ICLR 2025 LaTeX style guidelines, including incorrect margins.

These unresolved issues significantly limit the scope and potential impact of this work, reducing its appeal to the broader research community. Consequently, I believe it does not meet the high standards expected for acceptance at the prestigious ICLR conference.

---

### Decision · Program_Chairs · 2025-01-22

Reject